# Compliance to Guidelines in Prescribing Empirical Antibiotics for Individuals with Uncomplicated Urinary Tract Infection in a Primary Health Facility of Ghana, 2019–2021

**DOI:** 10.3390/ijerph191912413

**Published:** 2022-09-29

**Authors:** Helena Owusu, Pruthu Thekkur, Jacklyne Ashubwe-Jalemba, George Kwesi Hedidor, Oksana Corquaye, Asiwome Aggor, Allen Steele-Dadzie, Daniel Ankrah

**Affiliations:** 1Pharmacy Department, Korle Bu Teaching Hospital, Accra P.O. Box KB77, Ghana; 2Centre for Operational Research, International Union Against Tuberculosis and Lung Disease (The Union), 75001 Paris, France; 3The Union South-East Asia Office (The USEA), New Delhi 110016, India; 4Medwise Solutions, Nairobi P.O. Box 2356-00202 KNH, Kenya; 5World Health Organization Country Office, Roman Ridge, Accra P.O. Box MB142, Ghana; 6Polyclinic/Family Medicine Department, Korle Bu Teaching Hospital, Accra P.O. Box KB77, Ghana

**Keywords:** antimicrobial resistance, antimicrobial stewardship, audit-feedback mechanism, urinary tract infections, compliance, electronic medical records, sort it, operational research, Ghana, West Africa

## Abstract

Increasing trends in antimicrobial resistance among uropathogens call for rational use of empirical antibiotics for managing uncomplicated urinary tract infections (UTIs). In Ghana, standard treatment guidelines (STGs) for UTI recommend oral ciprofloxacin or cefuroxime for 5–7 days in females and 10–14 days in males. We conducted a cross-sectional audit using electronic medical records (EMR) to assess compliance to the STGs among adults (≥18 years) with uncomplicated UTIs diagnosed in a primary health facility between October 2019 and October 2021. Among 3717 patients, 71% were females and all had complete prescription details in the EMR. Of all the patients, 83% were prescribed empirical antibiotics, of whom 88% received oral ciprofloxacin or cefuroxime. Only 68% were prescribed antibiotics for the correct duration, which was significantly lower among males (10%) compared to females (90%). Among patients who received antibiotics, 60% were prescribed in line with the STGs. The results call for feedback to physicians about poor compliance to STGs with duration of antibiotic prescribed. Recommendations on 10–14 days duration of antibiotics for males needs to be reassessed and necessary amendments to STGs can be made. Leveraging the well-established EMR system, a real-time audit-feedback mechanism can be instituted to improve compliance with STGs.

## 1. Introduction

Urinary tract infections (UTIs) are among the most common infections worldwide, disproportionately affecting women [1]. Globally, an estimated 150 million people suffer from UTIs annually, contributing to about six billion US dollars in health costs [2]. Clinically, UTIs are classified as either uncomplicated or complicated [3,4]. Uncomplicated UTIs are those infections among non-pregnant individuals without any structural or functional abnormalities in the urinary tract [3]. Uncomplicated UTIs account for about 80% of all UTIs and are mostly (~75%) caused by *Escherichia coli* (*E. coli*) [3,5].

Individuals provisionally diagnosed with uncomplicated UTIs based on presenting symptoms and/or routine urine examination are treated with empirical antibiotics while awaiting urine culture and drug sensitivity test (CDST) results for confirmatory diagnosis and definitive treatment [5]. This is justifiable as recent meta-analyses have shown empirical antibiotics to be more effective than placebo or non-steroidal anti-inflammatory drugs in the early resolution of symptoms and in averting complications such as sepsis and pyelonephritis [6,7]. Moreover, in primary care settings of low-and-middle-income countries (LMICs) with poor access to urine CDST, empirical antibiotics form the mainstay of management.

Several countries have instituted standard treatment guidelines (STGs) for managing uncomplicated UTIs with recommendations on the dose and duration of empirical antibiotics [1,8]. Appropriate empirical antibiotics are recommended based on the antibiogram of uropathogens, costs, efficacy, and availability of the antibiotics in the country [5,9,10]. The use of inappropriate empirical antibiotics can prolong treatment with hospitalization, increase healthcare costs, and also contribute to the development of antimicrobial resistance (AMR) [11,12].

Adherence to guidelines while prescribing empirical antibiotics is crucial to reverse the increasing trends of AMR, including multi-drug resistance among uropathogens [13,14,15]. However, prescription audits from the United States, Sweden, the Netherlands, China, Tanzania, and Lebanon reported sub-optimal adherence to STGs while prescribing empirical antibiotics for uncomplicated UTIs [12,16,17,18,19,20,21,22,23]. Similar information on adherence to guidelines is scarce in primary care settings of LMICs, due to poor documentation of prescriptions and a lack of electronic databases for prescription audits [24]. The World Health Organization (WHO) recommends antimicrobial stewardship programmes in healthcare facilities of LMICs to establish either prospective (real-time) or retrospective prescription audits and feedback mechanisms to improve adherence to antimicrobial prescription guidelines [24].

In Ghana, a West African country, uncomplicated UTI is one of the major causes of outpatient visits [25]. A study conducted among inpatients with UTI in a tertiary teaching hospital of Ghana (2014) reported poor utilization (<20%) of CDST and a lack of subsequent switching of antibiotics based on the CDST results, underscoring the importance of effective empirical antibiotics for the management of UTI [26]. In 2017, the Ministry of Health of Ghana published the 7th edition of the STGs, which included the management of uncomplicated UTIs [27]. However, there has not been any audit to assess adherence to guidelines in primary care facilities, where most patients with uncomplicated UTIs are treated. This is mainly because of the lack of electronic medical records in the primary care settings [28].

The Korle Bu Polyclinic/Family Medicine Department (KBPFMD), a primary care facility, has established electronic medical records (EMRs) since October 2019. The diagnosis made and the drugs prescribed are captured in the EMRs for all the patients visiting the outpatient department. The availability of EMRs provided an opportunity for a prescription audit to assess adherence to management guidelines in the facility. The results from the prescription audit could be used to give feedback to treating physicians and act as a baseline for future serial audits to assess the utility of the audit-feedback mechanism for improving adherence to guidelines.

We, therefore, conducted a prescription audit of adults (aged ≥ 18 years) diagnosed with uncomplicated UTI in the outpatient departments of KBPFMD between October 2019 and October 2021 to determine: (1) the proportion prescribed empirical antibiotics and the distribution of antibiotics across the WHO AWaRE category, (2) the proportion prescribed empirical antibiotics as recommended in the STGs (dose, frequency, and duration), and (3) the patient and prescriber characteristics associated with not being prescribed empirical antibiotics as recommended in the STGs.

## 2. Materials and Methods

### 2.1. Study Design

This was a cross-sectional study using secondary data from the electronic medical records of the Korle Bu Polyclinic/Family Medicine Department.

### 2.2. Setting

#### 2.2.1. General Setting

Ghana is an LMIC having a population of about 30.8 million in 2021, with 51% females [29]. As of 2020, the estimated poverty rate in the country was 26% [30]. The Greater Accra Region, where the country’s capital (Accra) is located, is the most densely populated with about five million people [29].

In Ghana, most people (54%) access health care services at public health facilities managed by the Ghana Health Services and the Ministry of Health [31]. About 40% of all the health facilities in the country are private for-profit facilities located primarily in urban centers [32,33].The National Health Insurance Scheme (NHIS) was established in 2003 to avoid out-of-pocket expenditure for healthcare services and to achieve universal health coverage. The scheme supports out-patient and in-patient services, covering laboratory investigations and medicines [34]. As of 2021, the scheme covers about 66% of women and 53% of men in the country [35].

#### 2.2.2. Specific Setting

The KBPFMD is a 42-bed public health facility located in Accra. The facility is one of the ten satellite centers of the Korle Bu Teaching Hospital (KBTH), the largest teaching hospital in Ghana. The KBPFMD provides primary healthcare and emergency services to the people from the assigned sub-urban catchment area and KBTH staff and their dependents. The facility has a daily outpatient footfall of about 200 people, with three to six daily admissions. The health costs of most patients (about 90%) are covered under the NHIS or other private insurance schemes, and the rest pay out-of-pocket for the services.

The doctors, from the rank of medical officer to consultants, are involved in patient care at KBPFMD. The KBPFMD also serves as a training center for residents pursuing post-graduate specialization in family medicine and physician assistants.

#### 2.2.3. Management of Uncomplicated UTI

The KBPFMD, like all primary healthcare facilities in Ghana, is supposed to adhere to STGs for diagnosing and treating uncomplicated UTIs [27]. Patients with symptoms like frequent painful urination, cloudy or foul-smelling urine, suprapubic pain, and fever are suspected of UTI and referred to an in-house laboratory for routine urine examination. However, it is not uncommon for physicians to suspect and treat uncomplicated UTIs based solely on presenting symptoms without requesting routine urine examination. CDST is not usually requested by the physicians, as it is not available in the in-house laboratory, requiring patients to get it done in other laboratories with out-of-pocket payment and delays. Thus, patients with suspected uncomplicated UTIs are treated with empirical antibiotics and antipyretic drugs on an ambulatory basis.

The STGs recommend using oral ciprofloxacin or cefuroxime as an empirical antibiotic for the treatment of uncomplicated UTIs [27]. For adults, ciprofloxacin 500 mg every 12 h for 7 days in females and 10–14 days in males or cefuroxime 250–500 mg every 12 h for 5–7 days in females and 10–14 days in males is recommended [27]. Ciprofloxacin and cefuroxime recommended in the STGs are part of the essential drug list and are mostly available throughout the year. Individuals with complicated UTIs are usually admitted and treated with injectable antibiotics.

#### 2.2.4. Electronic Medical Records

In 2016, the Ministry of Health, in a bid to develop a centralized health data repository, began the rollout of eHealthcare services in public healthcare institutions in Ghana. The KBPFMD started using this electronic medical records system in October 2019. The socio-demographic details of the outpatients are captured upon registration. The clinician then assesses the patient and enters the clinical details into the EMR and makes a diagnosis according to the International Classification of Diseases-10 (ICD-10). Then, the clinicians make an electronic prescription detailing the drug (generic name), dosage, frequency, and duration for each medication prescribed. The electronic prescription is accessed at the pharmacy to dispense the prescribed medications to the patients. If some prescribed medications are unavailable, an electronic prescription is printed and issued to the patient for purchasing elsewhere. In addition, all the laboratory investigation results are available as scanned copies against the patient visit.

### 2.3. Study Population and Period

We included all the adults (aged ≥ 18 years) who were diagnosed with uncomplicated UTI (N39.0 of ICD-10) in the outpatient department of the KBPFMD between October 2019 and October 2021. Patients with complicated UTIs, those with urethral catheters, pregnant women, those with urologic abnormalities, and inpatients were excluded from the study.

### 2.4. Data Variables, Collection and Source of Data

Data were extracted from the EMR database of the KBPFMD. Adults diagnosed with uncomplicated UTIs during the study reference period were filtered from the database using a bespoke script run on the NodeJS framework. The extracted data included the patient, age, gender, visit date, NHIS status, occupation, diagnosis, and medications prescribed (including dosage, frequency, and duration). These data were then screened and filtered to remove all ineligible participants (repeat visits within 30 days of initial treatment). In addition, information on prescriber details (age, gender, and rank), routine urine examination (done/not done), and co-morbidities (diabetes and/or hypertension) were extracted manually from the scanned EMRs by two independent researchers since these were not part of the extracted data.

### 2.5. Data Analysis

The cleaned Microsoft Excel database was imported into Stata (version 16.0, Copyright 1985–2019, StataCorp LLC, College Station, TX, USA) for analysis. The patient and prescriber characteristics were categorized and summarized with frequencies and percentages.

For each study participant, we derived variables such as ‘prescription of any empirical antibiotic?’ (yes/no) and ‘prescription of empirical antibiotic drug recommended in the STG?’ (yes/no) based on the names of the drugs in the database. All those who were prescribed oral ciprofloxacin or cefuroxime were considered to have been prescribed empirical antibiotics recommended in the STGs. Similarly, we derived variables to code whether dose, frequency, and duration are as per the STG recommendations. We used the derived variables to create a composite variable called ‘prescribed empirical antibiotics in line with STGs’ (yes/no).

The proportion of patients with uncomplicated UTIs prescribed an empirical antibiotic and those prescribed empirical antibiotics in line with the STGs were presented as percentages with a 95% confidence interval (CI). A chi-square test was used to assess the difference in the proportion of prescribed empirical antibiotics across patient and prescriber characteristics. Frequency and percentages were used to describe the pattern of empirical antibiotics used across the WHO AWaRe categories.

We used log binomial regression for unadjusted analysis to assess the association of patient and prescriber characteristics with ‘prescription of empirical antibiotics not as recommended in STGs’. We conducted adjusted analysis using modified Poisson regression with all the patient and prescriber characteristics included. The prevalence ratios (PR) and adjusted prevalence ratios (aPR) with 95% CI were presented as the measure of association.

## 3. Results

### 3.1. Socio-Demographic and Clinical Characteristics

We included all 3717 adults diagnosed with uncomplicated UTIs in the outpatient department of the KBPFMD between October 2019 and October 2021. Of the 3717 patients included, the mean (standard deviation) age was 49.4 (19.5) years and 71% were females. Of the total, 79% had NHIS cards and 69% had undergone urine routine examination. Hypertension and diabetes were reported in 33% and 17% of the patients, respectively. For 44% of the patients, the treatment was prescribed by doctors of medical officer rank. (Table 1).

### 3.2. Prescription of Empirical Antibiotic

Of the 3717 patients included in the study, 3073 (83%, 95% CI: 81%–84%) were prescribed any empirical antibiotic. A lower proportion of patients who underwent routine urine examination (81%) received empirical antibiotics compared to those who did not (86%). None of the other patient and prescriber characteristics were significantly associated with the prescription of empirical antibiotics. (Table 1).

### 3.3. Prescription of Empirical Antibiotics as Recommended in STGs (Dose, Frequency and Duration)

Figure 1 summarizes the number and proportion of patients prescribed empirical antibiotics as recommended in the STGs. Of the 3073 patients prescribed any empirical antibiotic, 2714 (88%, 95% CI: 87–89%) were prescribed empirical antibiotics as recommended in the STGs (either oral ciprofloxacin or cefuroxime). Only one dose (stat dose) of injectable antibiotic (third-generation cephalosporin) was used in 4% of the patients as an empirical antibiotic.

Of the 2714 patients prescribed recommended empirical antibiotics, almost all (2712) were prescribed the antibiotic at the recommended dose and frequency. However, only 68% were prescribed for the duration as recommended in the STGs. Only 8% (62/777) of males were prescribed an empirical antibiotic for the recommended (10–14 days) duration, with the rest prescribed for less than ten days. In contrast, 92% (1784/1937) of the females received an empirical antibiotic for the recommended (5–7 days) duration.

Of the 3073 patients prescribed empirical antibiotics, 1847 (60%, 95% CI: 58–62%) were prescribed an empirical antibiotic at the dose, frequency, and duration as recommended in the STGs. About 9% of these patients were prescribed additional antibiotics concurrently.

### 3.4. Patient and Prescriber Characteristics Associated with Prescription of Empirical Antibiotics Not as Recommended in the STGs

On adjusted analysis, males (aPR-5.0, 95% CI: 4.6–5.5) compared to females and patients diagnosed in the year 2021 (aPR-1.2, 95% CI: 1.0–1.3) compared to those diagnosed in 2019 had a higher risk of being prescribed empirical antibiotics not as recommended in STGs. None of the other patient (age, gender, occupation, NHIS status, presence of comorbidities, and urine routine examination) and prescriber (gender and rank) characteristics were independently associated with being prescribed empirical antibiotics not as recommended in the STGs (Table 2).

### 3.5. Distribution of Prescribed Empirical Antibiotics across the WHO AWaRe Category

Of the 3378 empirical antibiotics prescribed for 3073 patients, 89% belonged to the WHO Watch category. In line with the STGs, cefuroxime (54%) and ciprofloxacin (31%) were the most common empirical antibiotics prescribed. Third-generation cephalosporins (cefixime and ceftriaxone) were prescribed in 2% of the patients. No antibiotics in the Reserve category were prescribed (Table 3).

## 4. Discussion

This first prescription audit from a primary healthcare facility in Ghana showed that a high proportion (90%) of the patients with uncomplicated UTIs were prescribed an empirical antibiotic. About 90% of the patients were prescribed an empirical antibiotic recommended in the STGs (either oral cefuroxime or ciprofloxacin). However, only two-thirds of the patients were prescribed empirical antibiotics for the recommended duration. This meant that only six out of ten patients were prescribed an empirical antibiotic at the dose, frequency, and duration in line with the STGs. Males had a five times higher risk of being prescribed empirical antibiotics not in line with the STGs, as less than 10% of the male patients received empirical antibiotics for more than ten days as recommended.

The findings of this study are important as it adds justification from Ghana for the WHO’s recommendation for embedding audit as a core component of the antimicrobial stewardship program in the health facilities of LMICs [24]. The audit-feedback system has been shown to effectively promote the rational use of antimicrobials and limit the emergence of AMR [36]. The recent revelation of five million annual global AMR deaths, with a notably higher risk of AMR deaths in Western Sub-Saharan African countries such as Ghana, makes it essential to tackle AMR swiftly [37].

The study has several strengths. First, we ascertained the adherence to STGs using a robust statistical programme in the Stata software, limiting the misclassification due to manual coding. Second, the large sample size enabled us to make precise estimates on adherence to STGs. Third, the EMR used for the study were complete without missing values for the key variables such as antibiotic prescribed, dose, frequency, and duration. Finally, we adhered to the STROBE (Strengthening the Reporting of Observational Studies in Epidemiology) guidelines for reporting the study findings [38].

The study has some limitations. First, we included all the uncomplicated UTI patients diagnosed in the facility and assumed all were prescribed an empirical antibiotic and not the definitive targeted antibiotic based on the CDST results. Thus, we might have underestimated the proportion of patients prescribed appropriate empirical antibiotics as those receiving targeted antibiotic based on CDST results would have been considered to have received empirical antibiotic not in line with the STGs. However, the impact of this limitation would be negligible since physicians mainly prescribed empirical antibiotics during the first patient visit. Second, as clinical (symptoms and their severity) and laboratory details could not be readily downloaded from the EMR, we failed to assess their association with adherence to STGs. Third, due to lack of CDST testing facilities at the KBPFMD, we were not able to present the antibiogram of the urine samples of the study participants. Fourth, we restricted the prescription audit to only empirical antibiotics and did not document the use of other drugs, including the switch of drugs based on CDST results. Finally, since this study was conducted only in a single primary care facility, the results cannot be generalized to other settings in the country.

Our study findings have useful implications for clinical practice in primary care settings. First, the use of empirical antibiotics in a high proportion (90%) of patients diagnosed with uncomplicated UTI at the KBPFMD is similar to studies from other LMICs [22,23,39]. This use of empirical antibiotics is justifiable as they are effective and form the mainstay of treatment in facilities with poor access to free CDST services [6,7]. However, to tackle AMR, it is vital to improve access to CDST and switch antibiotics when uropathogens are found to be resistant to empirical antibiotics [27]. Equipping the in-house laboratories with facilities for performing CDST and including these under the ambit of the NHIS would help to improve access and affordability for CDST. Future audits should assess the uptake of CDST and antibiotic switching when required.

Second, the high adherence (89%) to national STGs in terms of choice of recommended empirical antibiotic (oral cefuroxime or ciprofloxacin) for uncomplicated UTI in KBPFMD is appreciable. This is contrary to the findings from Tanzania (30%) and Lebanon (35%), where the physicians showed a relatively lower adherence to guidelines in the choice of empirical antibiotics for managing uncomplicated UTIs [22,23]. Similarly, a study from Ghana reported low (33%) adherence to STGs in the selection of antibiotics among primary care physicians for managing community-acquired pneumonia [40].

Previous studies reported poor knowledge about STGs among physicians in primary care settings and non-availability of the recommended drugs as primary reasons for low adherence in choosing the right antibiotic [22,23,39,41]. In contrast, treating physicians at KBPFMD are exposed to STGs during teaching-learning activities for postgraduate students (residents), as it is a satellite center of a teaching hospital. This highlights the importance of continuous medical education sessions for primary care physicians to improve their adherence to STGs. Moreover, uninterrupted availability of recommended antibiotics in the in-house pharmacy of KBPFMD and their availability on the NHIS medicines list could have positively contributed to high adherence to recommended empirical antibiotics.

Third, this study recorded a lower adherence (68%) to the recommended duration of prescribed antibiotics compared to adherence to the dose and frequency (~100%). Similarly, low adherence to prescribed recommended duration of treatment was reported in Australia, Lebanon, and the United States of America [20,22,42]. Taking antibiotics for longer than the recommended duration is linked with increased costs and side effects, while taking them for a shorter period could lead to AMR and recrudescence [43,44,45]. The availability of the oral cefuroxime and ciprofloxacin in packs of 10 or 14 makes it easy to prescribe a pack for five to seven days. It is also possible that prescribers are not mindful of the STG recommendation for a longer duration of treatment for males. This may explain the better adherence to the recommended duration for female patients (92%) who need to be prescribed empirical antibiotics for five to seven days. Whereas, in males who were supposed to be prescribed empirical antibiotics for 10–14 days, a prescription of one pack (maximum for seven days) would not be in line with the STGs. However, with recent evidence on the effectiveness of five to seven days of empirical antibiotic treatment among males, the Ministry of Health may consider revising the existing STGs [46,47].

Fourth, six out of ten (60%) patients had been prescribed the empirical antibiotic at a dose, frequency, and duration in line with the STGs. Though there are no standard benchmarks or targets, the 60% adherence to STGs in prescribing an empirical antibiotic for uncomplicated UTIs is commendable as it is higher than that found in many LMICs and some high-income countries (HICs) [20,22,23,39,41,42]. However, there is scope for improvement as some HICs, such as Switzerland, have achieved >85% adherence to treatment guidelines for the management of uncomplicated UTIs [48]. In the current study, the gap in adherence to STGs is mainly because of poor adherence to the recommended duration, especially among males. In this context, feedback to treating physicians in the facility about gaps in the duration of prescribed empirical antibiotics among males as well as qualitative determination of the causes would help to improve adherence to STGs.

Fifth, the use of additional antibiotics along with the recommended empirical antibiotics for uncomplicated UTIs was noted in the current study. Inappropriate prescribing of extra antibiotics will increase healthcare costs and promote AMR [49,50]. The unnecessary use of a single dose of intravenous antibiotics, especially a third-generation cephalosporin (ceftriaxone), is a worrying trend since these should be reserved for complicated UTIs [51]. Moreover, as third-generation cephalosporin-resistant *E. coli* are the leading cause of death among all the resistant pathogens, there is a need for judicious use of third-generation cephalosporins in uncomplicated UTIs mainly caused (75%) by *E. coli* [37]. The majority of antibiotics prescribed were from the WHO ‘Watch’ category since both recommended antibiotics fall in this category. It was commendable to note that none of the WHO Reserve category antimicrobials were used for treatment of uncomplicated UTIs.

Finally, the study showed that it is feasible to conduct a prescription audit using retrospective data collected in the EMRs. Thus, the audit can be replicated in other facilities with similar EMR systems. The information from this study can act as a baseline for serial audits to assess the impact of interventions aimed at improving adherence to STGs in a facility [24]. Moving forward, the KBPFMD should establish an audit-feedback system by incorporating an analytical package to the existing EMRs to derive indicators on adherence to STGs periodically. Instituting an antimicrobial stewardship program in the facility with a dedicated budget and personnel would be crucial to implementing the initiatives mentioned above to ensure the rational use of antibiotics as recommended in STGs [24].

## 5. Conclusions

The study from a primary health facility of Ghana showed that about nine in ten patients with uncomplicated UTIs were prescribed an empirical antibiotic, of whom 60% were as recommended in the STGs. The major gap was poor adherence in prescribing empirical antibiotics for a recommended duration of time, especially in male patients requiring more than ten days of antibiotics. Taking antibiotics for a shorter period could lead to AMR and recrudescence. However, it was appreciable that no antibiotic in the WHO ‘Reserve’ group was used for management of uncomplicated UTI. The study showed that it is feasible to conduct a prescription audit using the information in the EMRs. The pharmacy team should provide feedback on the major gap to the treating physicians and re-audit to assess the impact of such feedback. There is also scope for replicating this audit in other facilities to establish an audit-feedback mechanism to ensure rational use of antimicrobials.

## Figures and Tables

**Figure 1 ijerph-19-12413-f001:**
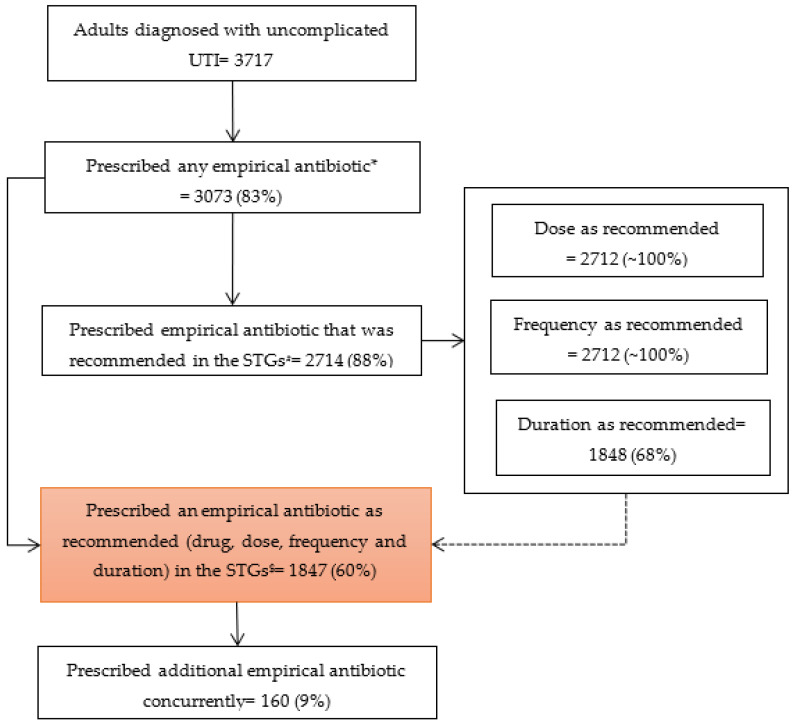
Flow-chart depicting empirical antibiotic prescription and adherence to the STGs among adults diagnosed with uncomplicated UTIs in the outpatient department of the Korle Bu Polyclinic/Family Medicine Department of Ghana during October 2019 to October 2021. * Prescription of antibiotic of any class by the treating physician; # Prescription of either oral ciprofloxacin or cefuroxime as recommended in the STGs; $ Prescription of either oral ciprofloxacin or cefuroxime recommended at the dose, frequency, and duration as recommended in the STGs; UTI = Urinary Tract Infection; STGs = Standard Treatment Guidelines.

**Table 1 ijerph-19-12413-t001:** Socio-demographic and clinical characteristics of adults with uncomplicated UTIs in the outpatient department of the Korle Bu Polyclinic/Family Medicine Department of Ghana during October 2019 to October 2021.

Characteristics	Total		Empirical Antibiotic Use		*p* Value ^e^
	N	(%) ^a^	n	(%) ^b^	
**Total**	3717	(100)	3073	(83)	
**Age in years**					
18–29	757	(20.4)	632	(83.5)	0.262
30–44	919	(24.7)	775	(84.3)	
45–59	804	(21.6)	661	(82.2)	
≥60	1237	(33.3)	1005	(81.2)	
**Gender**					
Male	1064	(28.6)	878	(82.5)	0.874
Female	2653	(71.4)	2195	(82.7)	
**Occupation**					
Employed	1909	(51.4)	1577	(82.6)	0.993
Unemployed	592	(15.9)	489	(82.6)	
Student	320	(8.6)	267	(83.4)	
Retired	238	(6.4)	195	(81.9)	
Not recorded	658	(17.7)	545	(82.8)	
**NHIS**					
Yes	2920	(78.6)	2402	(82.3)	0.202
No	797	(21.4)	671	(84.2)	
**Year**					
2019	644	(17.3)	533	(82.8)	0.255
2020	1746	(47.0)	1426	(81.7)	
2021	1327	(35.7)	1114	(84.0)	
**Comorbidities ^c^**					
Diabetes Mellitus	621	(16.7)	497	(80.0)	0.111
Hypertension	1231	(33.1)	1024	(83.2)	0.508
**Urine routine examination**					
Not done	1137	(30.5)	979	(86.1)	0.001
Done	2574	(69.3)	2089	(81.2)	
Missing	6	(0.2)	5	(83.3)	
**Prescriber gender**					
Male	2084	(56.0)	1719	(82.5)	0.915
Female	1623	(43.7)	1346	(83.0)	
Missing	10	(0.3)	8	(80.0)	
**Prescriber rank ^d^**					
Physician Assistant	36	(1.0)	27	(75.5)	0.570
Medical Officer	1616	(43.5)	1349	(83.5)	
Senior/Deputy Medical Officer	348	(9.4)	290	(83.3)	
Resident	1136	(30.6)	936	(82.4)	
Specialist	578	(15.6)	469	(81.1)	
Missing					

^a^ Column percentage with 3717 as the denominator; ^b^ Row percentage; ^c^ Multiple comorbidities are possible; ^d^ Those joining the service immediately after graduation are designated as medical officers. Based on their years of service, the medical officers progress through the ranks of the senior medical officer, deputy chief medical officer, and chief medical officer. Those doctors with postgraduate specialization join as senior residents and progress through the ranks of specialist, senior specialist, and consultant; ^e^ *p* Value based on the chi-square test; UTIs-Urinary Tract Infections; NHIS = National Health Insurance Scheme.

**Table 2 ijerph-19-12413-t002:** Patient and prescriber characteristics associated with prescription of empirical antibiotics not as recommended in the STGs among adults prescribed any empirical antibiotic for uncomplicated UTIs in the outpatient department of the Korle Bu Polyclinic/Family Medicine Department of Ghana during October 2019 to October 2021.

Characteristics	Total	Empirical Antibiotics not as Recommended in STGs		Unadjusted ^b^		Adjusted ^c^	
	N	n	(%) ^a^	PR	(95% CI)	aPR	(95% CI)
**Total**	3073	1226	(39.9)				
**Age in years**							
18–29	632	234	(37.0)	1		1	
30–44	775	313	(40.4)	1.1	(1.0–1.3)	1.1	(1.0–1.2)
45–59	661	251	(38.0)	1.0	(0.9–1.2)	1.0	(0.9–1.2)
≥60	1005	428	(42.6)	1.1	(1.0–1.3)	1.0	(0.9–1.2)
**Gender**							
Male	878	819	(93.3)	5.0	(4.6–5.5)	5.0	(4.6–5.5) ^f^
Female	2195	407	(18.5)	1		1	
**Occupation**							
Employed	1577	632	(40.1)	0.9	(0.9–1.1)	1.0	(0.8–1.1)
Unemployed	489	200	(40.9)	1		1	
Student	267	99	(37.1)	0.9	(0.8–1.1)	0.9	(0.8–1.2)
Retired	195	96	(49.2)	1.2	(1.0–1.5)	1.0	(0.8–1.1)
Not recorded	545	199	(36.5)	0.9	(0.8–1.1)	0.9	(0.8–1.0)
**NHIS**							
Yes	2402	938	(39.1)	1		1	
No	671	288	(42.9)	1.1	(1.0–1.3)	1.0	(0.9–1.1)
**Year**							
2019	533	200	(37.5)	1		1	
2020	1426	565	(39.6)	1.0	(0.9–1.2)	1.0	(0.9–1.1)
2021	1114	461	(41.4)	1.1	(1.0–1.3)	1.2	(1.0–1.3) ^f^
**Comorbidities ^d^**							
Diabetes Mellitus	497	197	(39.6)	1.0	(0.8–1.1)	1.0	(0.9–1.1)
Hypertension	1024	412	(40.2)	1.0	(0.9–1.1)	1.0	(0.9–1.1)
**Urine routine examination**							
Done	2089	867	(41.5)	1		1	
Not done	979	356	(36.4)	0.9	(0.8–1.0)	0.9	(0.8–1.0)
Not recorded	5	3	(60.0)	1.5	(0.7–3.3)	1.3	(0.7–2.3)
**Prescriber gender**							
Male	1719	708	(41.2)	1		1	
Female	1346	516	(38.3)	0.9	(0.9–1.0)	0.9	(0.9–1.0)
Missing	8	2	(25.0)	0.6	(0.2–2.0)	0.5	(0.1–1.9)
**Prescriber rank ^e^**							
Physician Assistant	27	11	(40.7)	0.8	(0.5–1.3)	1.0	(0.6–1.5)
Medical Officer	1349	545	(40.4)	0.9	(0.8–1.0)	1.0	(0.9–1.1)
Senior/Deputy Medical Officer	290	133	(46.0)	1		1	
Resident	936	367	(39.2)	0.8	(0.7–1.0)	1.0	(0.9–1.1)
Specialist	469	169	(36.0)	0.8	(0.6–0.9)	0.9	(0.7–1.0)
Missing	2	1	(50.0)	0.9	(0.2–4.3)	1.3	(0.4–4.8)

^a^ Row percentage; ^b^ log binomial regression; ^c^ modified Poisson regression with all the patient and prescriber characteristics included; ^d^ not having the specific comorbidity is the reference; ^e^ Those joining the service immediately after graduation are designated as medical officers. Based on their years of service, the medical officers progress through the ranks of the senior medical officer, deputy chief medical officer, and chief medical officer. Those doctors with postgraduate specialization join as senior residents and progress through the ranks of specialist, senior specialist, and consultant; ^f^ statistically significant (*p* value < 0.05); UTI = Urinary Tract Infection; STGs = Standard treatment guidelines; PR = Prevalence Ratio; CI = Confidence Interval; aPR = Adjusted Prevalence Ratio.

**Table 3 ijerph-19-12413-t003:** Distribution of empirical antibiotics across WHO AWaRe categories prescribed for adults diagnosed with uncomplicated UTIs in the outpatient department of the Korle Bu Polyclinic/Family Medicine Department of Ghana during October 2019 to October 2021.

Antibiotic	N	(%)
Total *	3378	(100) #
** *Access* **		
Tinidazole	119	(3.5)
Nitrofurantoin	90	(2.7)
Doxycycline	88	(2.6)
Amoxicillin/clavulanic acid	52	(1.5)
Secnidazole	20	(0.6)
Metronidazole	8	(0.2)
Clindamycin	2	(0.06)
Amoxicillin	1	(0.03)
Sulfamethoxazole/trimethoprim	1	(0.03)
** *Watch* **		
Cefuroxime	1831	(54.2)
Ciprofloxacin	1036	(30.7)
Cefixime	38	(1.1)
Ceftriaxone	33	(1.0)
Levofloxacin	31	(0.9)
Azithromycin	23	(0.7)
Cefpodoxime	3	(0.1)
Clarithromycin	2	(0.1)

* Total number of antibiotics used among the 3073 uncomplicated UTI patients who were prescribed empirical antibiotics; # Column percentage with total number of antibiotics as the denominator; UTI = Urinary Tract Infection; AWaRe = Access, Reserve, and Watch; WHO = World Health Organization.

## Data Availability

The dataset used in this paper has been deposited at https://doi.org/10.6084/m9.figshare.20418849.v1 and is available under a CC BY 4.0 license.

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
