# Peer review of "Compliance to Guidelines in Prescribing Empirical Antibiotics for Individuals with Uncomplicated Urinary Tract Infection in a Primary Health Facility of Ghana, 2019–2021"

_ijerph, 2022, doi:10.3390/ijerph191912413_

Round 1
Reviewer 1 Report
1) General comments
The authors conducted a cross-sectional audit using electronic medical records (EMR) to assess compliance to the standard treatment guidelines (STG) among adults (≥18 years) with uncomplicated urinary tract infections (UTIs) diagnosed in a primary health facility between October 2019 and October 2021 in Ghana. About nine in ten patients with uncomplicated UTIs were prescribed an empirical antibiotic, of whom 60% were as recommended in the STGs. The major gap was poor adherence in prescribing empirical antibiotics for a recommended duration of time, especially in male patients requiring more than ten days of antibiotics.
The reviewer generally agrees with the conclusion and the importance of this study.
However, there are several issues need to improve. The reviewer would like suggests several issues as follows;
2) Specific comments for revision
a) Major
#1 Please clarify the diagnostic criteria for uncomplicated UTIs.
Are UTIs based on clinical symptoms alone without urine examination?
#2 Please add the urine culture and drug sensitivity test (CDST) to the limitation since the CDST is not easily available due to health care system.
Minor
#1 There are too many paragraphs in this paper and it would be better to summarize them.
#2 Please add the data that led to the recommendation of oral ciprofloxacin or cefuroxime and the duration of prescription for uncomplicated UTIs in STG.
Author Response
"Please see the attachment"

Reviewer 2 Report
Dear Authors,
Congratulations on your study, it was a pleasure to read and review the article.
Of course there are some minor corrections which I suggest:
Strengths and limitations:
The first and the most important limitation of the study is a single-centre setting. Thus, any generalization of the results needs to be made very carefully, and I would suggest it even in the conclusion section. Secondly, the percentages of patients who are treated adequately may vary hugely, you need to underline it; those are (only) the reuslts from your hospital. Another interesting option is a try to reach those patients in order to fllow-up the effects of the treatment, and although it is time-consuming and it was not the aim of the study, you could try to do such an analysis in order to start a small revolution in the local guidance, showing for example that a shorter period of treatment does not result in long-term sequelae.
Best regards
Author Response
"Please see the attachment."

Reviewer 3 Report
This is a well written paper presenting important findings to public health. However, prior the paper is published I strongly recommend to introduce the following adjustments:
1. I would expect to see deeper discussion of Table 3 results, with some indications provided also in Conclusions section.
2. Please move strengths and limitations of the study to the end of Discussion section. Among limitations of the study please note and comment the regional scope of the studied sample.
3. Conclusions section requires major improvements. Please provide consequences of wrong duration (see line 352-354) and implications of obtained results to various stakeholders. Now, the results are presented too poorly.
Author Response
"Please see the attachment."
